Evolutionary couplings detect side-chain interactions

http://orcid.org/0000-0001-9476-0104 Hockenberry Adam J. adam.hockenberry@utexas.edu
http://orcid.org/0000-0002-7470-9261 Wilke Claus O.
Department of Integrative Biology, The University of Texas at Austin , Austin, TX , USA
Gelfand Mikhail
Electronic publication date: 2019 Jul 8
Publication date: 2019
Volume: 7
Electronic Location ID: e7280
Received 2019 Mar 14; Accepted 2019 Jun 9
Copyright: © 2019 Hockenberry and Wilke
Copyright year: 2019
Copyright holder: Hockenberry and Wilke
License: This is an open access article distributed under the terms of the Creative Commons Attribution License, which permits unrestricted use, distribution, reproduction and adaptation in any medium and for any purpose provided that it is properly attributed. For attribution, the original author(s), title, publication source (PeerJ) and either DOI or URL of the article must be cited.
License URL: https://creativecommons.org/licenses/by/4.0/

Keywords: Protein evolution, Contact prediction, Evolutionary couplings, Structural constraints, Epistasis

Funding: National Science Foundation Cooperative Agreement DBI-0939454 (BEACON Center) The National Institutes of Health R01 GM088344 This work was funded by the National Science Foundation Cooperative Agreement no. DBI-0939454 (BEACON Center) and the National Institutes of Health grant R01 GM088344. The funders had no role in study design, data collection and analysis, decision to publish, or preparation of the manuscript.

==============================
Patterns of amino acid covariation in large protein sequence alignments can inform the prediction of de novo protein structures, binding interfaces, and mutational effects. While algorithms that detect these so-called evolutionary couplings between residues have proven useful for practical applications, less is known about how and why these methods perform so well, and what insights into biological processes can be gained from their application. Evolutionary coupling algorithms are commonly benchmarked by comparison to true structural contacts derived from solved protein structures. However, the methods used to determine true structural contacts are not standardized and different definitions of structural contacts may have important consequences for interpreting the results from evolutionary coupling analyses and understanding their overall utility. Here, we show that evolutionary coupling analyses are significantly more likely to identify structural contacts between side-chain atoms than between backbone atoms. We use both simulations and empirical analyses to highlight that purely backbone-based definitions of true residue–residue contacts (i.e., based on the distance between Cα atoms) may underestimate the accuracy of evolutionary coupling algorithms by as much as 40% and that a commonly used reference point (Cβ atoms) underestimates the accuracy by 10–15%. These findings show that co-evolutionary outcomes differ according to which atoms participate in residue–residue interactions and suggest that accounting for different interaction types may lead to further improvements to contact-prediction methods.

Introduction

A long-standing problem in physical biology is to predict the structure of a protein based solely on its amino acid sequence (Anfinsen, 1973; Sadowski & Jones, 2009; Marks, Hopf & Sander, 2012). Despite advances in x-ray crystallography (Miao et al., 2015; Batyuk et al., 2016), NMR spectroscopy (Liu et al., 2005; Denisov & Sligar, 2016), cryo-electron microscopy (Liao et al., 2013; Amunts et al., 2014; Punjani et al., 2017), and template-based homology modeling (Söding, 2005; Biasini et al., 2014), the pace at which new structural data is generated pales in comparison to the rate at which we are accumulating new genomes and gene sequences (Rose et al., 2017). As a result, many protein families still lack even a single representative structure or set of functional annotations (Wyman et al., 2018).

In recent years, however, several groups have made substantial improvements to de novo structure prediction by leveraging co-evolutionary information contained within large sequence alignments of protein homologs (Lapedes et al., 1999; Burger & Van Nimwegen, 2008, 2010; Marks et al., 2011; Morcos et al., 2011; Sulkowska et al., 2012; Kamisetty, Ovchinnikov & Baker, 2013). One key result has been that residues that co-evolve with one another frequently do so as a result of their spatial proximity within the protein structure, that is, a mutation to one site can be compensated for by subsequent mutations to other directly interacting sites (Göbel et al., 1994; Shindyalov, Kolchanov & Sander, 1994; Weigt et al., 2009). By determining an “evolutionary coupling” score for all pairs of amino acid residues within a sequence alignment—and assuming that the highest-scoring residue–residue pairs are in close spatial proximity within the structure—the search space of computational protein folding methods can be constrained, resulting in accurate 3D-structure determination (Marks et al., 2011; Hopf et al., 2012; Ovchinnikov et al., 2017). Other applications have used evolutionary coupling scores to predict protein binding partners and interfaces (Burger & Van Nimwegen, 2008; Hopf et al., 2014; Ovchinnikov, Kamisetty & Baker, 2014), as well as to predict the effect of mutations on protein stability and function (Hopf et al., 2017). Many of these approaches have been further improved through the use of machine learning (Cheng & Baldi, 2007; Jones et al., 2015; Michel et al., 2017), and specifically deep neural networks that leverage evolutionary couplings along-side numerous other protein features (Tegge et al., 2009; Di Lena, Nagata & Baldi, 2012; Xiong, Zeng & Gong, 2017; Stahl, Schneider & Brock, 2017; He et al., 2017; Wang et al., 2017; Riesselman, Ingraham & Marks, 2018; Liu et al., 2018; Wozniak et al., 2018; Jones & Kandathil, 2018; Adhikari, Hou & Cheng, 2018; Hanson et al., 2018).

During the development and testing of evolutionary coupling algorithms, predictions of residue–residue couplings from large multiple-sequence alignments are frequently benchmarked against a set of known protein structures to determine their accuracy in identifying residue–residue contacts (Figliuzzi, Barrat-Charlaix & Weigt, 2018; Schaarschmidt et al., 2018; Wang, Sun & Xu, 2018). The large number of protein structures that have been solved at atomic resolution provides a training dataset where intramolecular contacts are known (Rose et al., 2017). However, even the most high-resolution crystal structures of proteins require extrapolation from the location of particular atoms and residues to classify residue–residue contacts (Seeliger & De Groot, 2007; Sathyapriya et al., 2009; Duarte et al., 2010; Yuan, Chen & Kihara, 2012). A commonly used heuristic is to determine that any amino acid residue that lies within some pre-defined physical distance—frequently 8Å—of another amino acid residue is said to be in structural contact (Marks et al., 2011).

Different reference points can be used when determining the distance between any pair of amino acid residues, and prior research has shown that the choice of reference point can matter in some contexts. For example, in the context of structural determinants of sequence conservation, it has been found that sequence conservation is more strongly correlated to the number of residue–residue contacts identified via side-chain centers than identified via Cα atoms (Lin et al., 2008; Marcos & Echave, 2015; Shahmoradi & Wilke, 2016). However, it is unknown whether the choice of reference point is important for evaluating and interpreting modern evolutionary coupling approaches.

Additionally, some studies eschew fixed reference points entirely. Residue–residue contacts can be defined according to whether the minimum distance between all possible pairs of heavy atoms between two residues falls below a set distance cutoff (Ovchinnikov et al., 2017). However, it is unclear whether all atoms should be considered in this calculation or whether backbone and side-chain atoms should be treated differently. Further, there are a number of ever more complex approaches that can be used to define contacts given a protein structure, including the use of hydrogen bonding and residue interaction networks (Doncheva et al., 2011; Bhattacharya & Cheng, 2013; Piovesan, Minervini & Tosatto, 2016), as well as correlated residue movements from molecular dynamics simulations (Scarabelli & Grant, 2013; Doshi et al., 2016; Serçinoğlu & Ozbek, 2018).

Currently, there are no accepted standards in the field for how to determine a set of residue–residue contacts for a given protein structure. Further, there has yet to be a systematic analysis of whether co-evolutionary signatures are more- or less-closely related to different types of intramolecular contacts that may exist within a given structure. Here, we systematically test the accuracy of several evolutionary coupling algorithms against true positive contacts defined via Cα, Cβ, or side-chain geometric centers, as well as via minimum distances between all atoms or atoms residing in side chains only. We find that residue–residue contacts defined according to side-chain centers and side-chain atoms are much more accurately predicted by evolutionary couplings. These results imply that the dominant epistatic effects resulting in co-evolutionary signatures arise from interactions between side-chain atoms. Our findings highlight the importance of the choice of contact definition and provide insight into the constraints governing the evolution of protein structures.

Materials and Methods

Dataset compilation and processing

We downloaded the so-called PSICOV dataset of 150 proteins that have been extensively studied (Jones et al., 2012, 2015; Jones & Kandathil, 2018). We processed each starting PDB file to select a single chain, ensure a consistent numbering of residues (1...n), test for unknown or non-standard residues, select the most likely state for all disordered sequence atoms, and remove all extraneous information (including “HETATM” lines). Next, to ensure that all residues were represented in full and repair those that were not, we used PYROSETTA to read in the “.PDB” files using the “pose_from_pdb” function and wrote the output as our final clean structure (Chaudhury, Lyskov & Gray, 2010).

Determining structural contacts and contact-types

From each cleaned “.PDB” file, we calculated residue–residue distance matrices using custom python scripts (the Euclidean distance from three-dimensional atomic coordinates). All residues contain a Cα atom so this calculation was straightforward. For Cβ calculations, we used the Cβ atom of all residues except glycine, for which we continued to use the Cα atom. For side-chain center calculations, we calculated the geometric center of each residue based on the coordinates of all non-backbone heavy atoms. This calculation included Cβ atoms but excluded Cα atoms for all amino acids except glycine, for which we again continued to use Cα as the reference point.

To calculate minimum atomic distances between two residues, we calculated all pairwise Euclidean distances between heavy atoms and selected the minimum distance. In extending this analysis to only consider side-chain atoms, we continued to consider Cβ atoms as part of the side chain but not Cα. Again, we relaxed this restriction for glycine and included Cα as a side-chain atom to permit calculations for this amino acid.

For all methods, contacts were assessed by first removing all residue–residue pairs where the two residues were shorter than 12 amino acids apart in chain distance. Contacts were determined throughout this manuscript for each structure according to an 8Å cutoff between Cα atoms. Since accuracy values are partially dependent on the number of true positives that are called, we maintained a constant number of true positive contact classifications throughout to facilitate comparison between methods. For each contact definition (Cβ, side-chain center, minimum atomic distances), we selected n residue–residue pairs with the shortest distances where n is the number of contacts defined according to the aforementioned Cα-based method. Thus, the actual distance cutoff used to classify contacts for these other metrics varies slightly from protein to protein.

To classify residue–residue pairs (a and b) via their side-chain orientations, we chose a reference residue (a) and drew two vectors: (i) from the Cα atom coordinate to the side-chain center for that residue and (ii) from the Cα atom coordinate to the Cα atom coordinate for the other residue in question (b). If the angle between these two vectors was less than π/2 radians, the side chain of residue a was said to point toward residue b. To determine the residue’s classification we next repeated the calculation using residue b as the reference and finally classified the residue–residue pair accordingly.

Evolutionary coupling algorithms

For each of the 150 proteins in our dataset, we followed a principled method to retrieve homologous sequences. We first extracted the amino acid sequence from the “.PDB” file. We next used PHMMER to search through progressively larger databases in order to retrieve up to 10,000 homologous sequences (Potter et al., 2018). To do so, we downloaded local versions of the rp15, rp35, rp55, and rp75 databases (Chen et al., 2011). We first searched the smallest, least redundant, database for each sequence using an E-value threshold of 0.0001. For any sequence with greater than 10,000 hits we stopped and selected the top scoring 10,000 hits for further analysis. For sequences with fewer than 10,000 hits we moved to the next largest database and repeated the process. Finally for the small number of sequences for which we did not accumulate at least 1,000 sequences in the largest database (rp75), we used the online version of PHMMER to search the UniprotKB database and downloaded the maximum results.

For each protein, we next aligned the hits along-side the reference sequence using MAFFT (v7.380, L-INS-i method with default parameters) (Katoh & Standley, 2013). Next, we cleaned these results to remove all columns that were gapped in the reference (“.PDB”) sequence. All other columns and sequences in the sequence alignments were retained regardless of gap coverage.

Using these alignments, we next calculated evolutionary couplings between residue–residue pairs. All results in the main manuscript are displayed using CCMpred with default parameters (0.8 local sequence re-weighting threshold, 0.2 pairwise regularization coefficients, average product correction) (Seemayer, Gruber & Söding, 2014). We additionally used the “plmc” method from the EVcouplings framework with default parameters (no average product correction) (Hopf et al., 2018) and PSICOV (default parameters excepting: “−z 50 −r 0.001”) (Jones et al., 2012) to ensure the robustness of our findings.

Except where otherwise noted, main text results were calculated using the top L/2 couplings for each protein where L is the length of the reference amino acid sequence. Positive predictive value (PPV) is calculated as the number of classified contacts among these top couplings divided by the total number of top couplings considered.

Evolutionary simulations

For the example protein used throughout the text (PDB:1AOE) we performed mutation accumulation simulations using PYROSETTA (Chaudhury, Lyskov & Gray, 2010). We first read in the “.PDB” structure (with disulfide bonds disabled), and minimized it to optimize thermodynamic stability via rotamer selection and backbone movements. We next fixed the backbone, and implemented an expedited evolutionary algorithm to select amino acid point mutations (no insertions or deletions were allowed) according to their predicted impact on structural stability (Teufel & Wilke, 2017; Jiang et al., 2018). At each step, we selected a random amino acid position, and attempted to mutate it randomly to one of the remaining 19 amino acids. We re-packed the structure within a 12Å radius and determined whether or not to accept the mutation based off of the resulting change in structural stability. Mutations which either did not alter or increased stability (i.e., resulted in a decreased ΔG) were accepted with a probability of 1. Mutations that decreased stability were accepted with a probability proportional to their ΔΔG as in Teufel & Wilke (2017). At the end of the evolutionary process, the resulting amino acid sequence was stored for future analysis.

We performed thousands of independent replicates of this expedited evolutionary process where we altered the number of accepted mutations that we accumulated before halting the simulation, the number of replicate evolutionary experiments that we performed, and the fraction of the initial wild-type stability value that we used for our selection criteria. Collections of the resulting sequences were analyzed via evolutionary coupling algorithms in the same manner as empirical sequences, with no need for sequence alignment.

Results

Structural contact definitions

Putatively true interactions between amino acid residues within a given protein family are frequently derived from the distance between residues in a representative protein structure. Figure 1 depicts an example protein structure (PDB:1AOE) alongside two symmetric matrices depicting all residue–residue distances (in angstroms, Å) defined according to either the distance between the Cα atoms of individual residues or the distance between the geometric centers of each residue’s side chain. We use the geometric center of a residue’s side chain throughout this manuscript and note that, unlike the center-of-mass, this metric includes only information about the spatial coordinates of atoms and is thus agnostic to their identity/mass.

Figure 1 Constructing contact maps from protein structures.

(A) An example structure (PDB:1AOE). (B) A symmetric distance matrix between all pairs of amino acid residues measured from each residue’s Cα atom. (C) Medium- to long-range contacts (≥12 residues apart along the linear chain are identified using an 8Å cutoff (dark blue). (D) and (E) Same methodology as depicted in (B) and (C), using the geometric center of each residue’s side chain as a reference point for measuring distances.

From these residue–residue distance matrices, we created contact matrices by using an 8Å, Cα-based distance cutoff to define contacts for a given protein. To facilitate a direct comparison between methods, we use the same absolute number of contacts to determine a comparable distance cutoff (specific to each protein) to use for side-chain center distances such that an equal number of true contacts were identified regardless of the reference point used to measure distances. For instance, in Fig. 1 we found that the 8Å, Cα-based distance cutoff resulted in 295 contacts. We thus chose the distance cutoff for side-chain centers such that 295 contacts were identified, corresponding to a distance cutoff of 7.33Å for this example protein. Additionally, for most applications the structurally interesting contacts are those of amino acids that are not close to each other in the linear sequence. We define those here as amino acid pairs separated by a chain distance of at least 12 residues, and we only consider this subset of possible contacts for this example and the remainder of this manuscript.

While the distance matrices look similar for an example protein when calculated via Cα atoms or side-chain centers (Figs. 1B compared to 1D), the resulting maps of residue–residue contacts show considerable heterogeneity (Figs. 1C compared to 1E). More quantitatively, the set of all residue–residue distances measured by Cα atoms are highly correlated with comparable distances measured via side-chain centers (Fig. 2A). However, this strong overall correlation obscures important differences in contact definitions which we observe when focusing within the narrow region where direct amino acid residue contacts are defined (Fig. 2B). For 1AOE, we identified a total of 295 contacts according to the 8Å Cα-based distance cutoff. Of the shortest 295 contact distances identified via side-chain centers, the percent of residue–residue pairs that appear in both definitions is only 56%.

Figure 2 Relationship between different contact identification methods.

(A) Correlation between residue–residue distances in PDB:1AOE measured according to Cα atoms and side-chain centers. (B) A zoomed in view (right) highlights variably defined residue–residue contacts indicated by the various colors. Cutoffs for defining contacts are 8 and 7.33Å for the Cα and side-chain center based metrics for this protein. (C) Distribution of Spearman’s correlation coefficient values (ρ) between residue–residue distances for 150 different proteins. (D) Distribution of the percent agreement for contact definitions for the same set of proteins (Fig. S2 shows a comparable comparison between Cβ and side-chain center distances).

To assess the generality of these findings, we applied this analysis to a commonly used benchmark set of 150 proteins (Jones et al., 2012, 2015; Jones & Kandathil, 2018). Across all of these proteins, we observed a median correlation of 0.97 between residue–residue distances calculated via Cα atoms and side-chain centers (Fig. 2C), but a median overlap of just 63% between contacts defined via Cα and side-chain centers (Fig. 2D). The distribution of protein-specific distance cutoffs that we used to identify equal numbers of contacts across methodologies was approximately normal and in the range of 6–8 Ås (Fig. S1).

We focused on comparing the use of Cα atoms and side-chain centers as reference points to determine contacts, since these two features represent extreme ends of the spectrum. In practice, Cβ atoms occupy an intermediate location between the backbone and the side-chain center and are one of the most frequently used reference points (Seeliger & De Groot, 2007; Sathyapriya et al., 2009; Duarte et al., 2010; Yuan, Chen & Kihara, 2012). We repeated the primary results of this section by comparing the use of Cβ atoms and side-chain centers as reference points and observed only a 78% overlap in residue–residue contacts identified by these two methods (Fig. S2). Together, these findings highlight that true contacts vary substantially according to the reference point used to measure residue–residue distances.

Evolutionary couplings reflect side-chain contacts in simulated sequence alignments

While the previous analysis of empirical structures shows that the choice of reference point has important consequences for true contact identification, it is not clear which of the different methods is more biologically correct or practically meaningful. We thus turned our attention to a simplified biophysical system of simulated protein evolution to test the ability of evolutionary coupling analyses to recover intramolecular contacts. We used the ROSETTA modeling software (Chaudhury, Lyskov & Gray, 2010; Leaver-Fay et al., 2011; Kellogg, Leaver-Fay & Baker, 2011) to perform all-atom simulations of the evolutionary process (Teufel & Wilke, 2017; Jiang et al., 2018) while selecting for the maintenance of protein stability (expressed as a fraction of the initial PDB model stability). We simulated thousands of independent evolutionary trajectories and used the resulting amino acid sequences from these simulations to calculate evolutionary couplings using CCMpred with default parameters (Seemayer, Gruber & Söding, 2014). Within this defined system, we are able to remove the constraints of phylogenetic biases, limited data availability, homo-oligomerization, insertions/deletions, and changes in evolutionary pressures over time between species—all of which partially limit the power of algorithms to detect true evolutionary couplings in real data (Anishchenko et al., 2017).

We continued to use 1AOE as an example structure and varied several parameters of our simulation to ensure robust results. We defined true positive residue–residue contacts according to the original PDB structure using residue–residue distances calculated between different quantities for comparison (Cα, Cβ, and side-chain center with respective distance cutoffs of 8, 7.5, and 7.33Å). To assess the accuracy of evolutionary couplings, we determined the PPV of the top L/2 couplings—where L is the chain length of the protein under investigation, L = 192 in the case of 1AOE. For Cα-based contact definitions, we found that the PPV increases rapidly according to the number of independent sequences that we simulated and consequently used as input for evolutionary coupling analyses (Fig. 3A). For these simulations, we ran each independent evolutionary simulation until we accepted a number of point mutations totaling 10 times the length of the protein sequence. However, regardless of the selection strength that we imposed on the sequence evolution (colored lines in Fig. 3A), PPV values plateaued at a value below 0.75. This indicates that more than a quarter of the top ranked evolutionary couplings represent incorrect predictions and are not true intramolecular contacts. By contrast, when we analyzed the same evolutionary coupling values but instead used side-chain center distances to define true contacts, we observed that nearly all of the top ranked couplings were defined as true contacts (PPV values approached 1).

Figure 3 Comparing simulation-derived evolutionary couplings to different contact definitions.

(A) For each of five separate selection strengths (colored lines), we ran evolutionary simulations until a number of mutations totaling 10 times the length of the protein were accumulated per replicate. We varied the number of independent replicate sequences (x-axis) that were used as input for evolutionary coupling analysis, and found that the resulting evolutionary couplings fail to fully recover Cα defined contacts (8Å) for PDB:1AOE. (B) By contrast, contacts defined via side-chain centers (7.33Å) are near-perfectly recovered across a range of simulation parameters. (C) and (D) Similar to parts (A) and (B), but along the x-axis we now show results from simulations where a different number of accepted mutations were accumulated per sequence. We fixed the number of replicate sequences that were simulated—and used as input for evolutionary coupling analysis—at 3,000 for each of these data points (Results comparing Cβ and side-chain center contact definitions, can be found in Fig. S3).

We additionally explored how the number of mutations accumulated per sequence affected the ability of evolutionary coupling algorithms to recover intramolecular contacts. We fixed the number of replicate sequences at 3,000, and found that PPV values showed minimal variation according to the number of accepted mutations that had accumulated per sequence over the course of our in silico evolution (Fig. 3C). As before, however, prediction accuracies were substantially higher when we defined true contacts according to side-chain center distances (Fig. 3D). These simulation results highlight that—across numerous parameter combinations—the top L/2 evolutionary couplings corresponded to true intramolecular contacts as long as true positives were defined according to side-chain centers and not Cα carbons. Definitions of contacts based on Cβ atoms resulted in intermediate accuracy, plateauing at higher values than Cα but lower than side-chain centers (Fig. S3).

Evolutionary couplings reflect side-chain contacts in natural sequence alignments

To see how evolutionary couplings compare to different definitions of true residue–residue contacts in empirical data, we used PHMMER to identify sequence homologs for each of the 150 proteins (see Materials and Methods for details). We assessed the relationship between evolutionary couplings and structural contacts for all proteins by calculating the PPV of the highest L/2 couplings.

As expected, the PPV between the top L/2 evolutionary couplings and Cα-based contacts varied substantially across the 150 structures. This variation may result from a number of different effects, and we observed a clear correlation between PPV values and the number of homologous sequences used to determine evolutionary couplings (Fig. S4). Despite the variability in prediction accuracy between proteins, we observed systematic differences in the PPV according to which metric was used to identify true positive contacts from the PDB structure files (Fig. 4A). When compared to Cα based distances, residue–residue distances measured according to Cβ atoms resulted in significantly higher PPVs, and side-chain based contact distances resulted in even further improvements (Wilcoxon signed-rank test, p < 10−20 for all comparisons). Further, the magnitude of the effect was substantial: across all 150 proteins the median percent increase in PPV between Cα and side-chain center contact identification methods was 43% (Figs. 4B and 4C). Even between the more similar Cβ and side-chain center methods, the median percent increase in accuracy was 13% (Figs. 4D and 4E). Both comparisons were highly significant and persisted across the entire range of PPVs represented within our dataset (Figs. 4B and 4D). Additionally, these results were highly consistent across different evolutionary coupling algorithms (Figs. S5 and S6).

Figure 4 Accuracy of evolutionary couplings derived from empirical alignments.

(A) For a diverse set of 150 proteins, the PPV of the top L/2 evolutionary coupling scores—derived from empirical sequence alignments—is progressively higher when intramolecular contacts are defined according to Cα atoms, Cβ atoms, and side-chain centers. Similarly, PPVs are higher when computing contacts based on side-chain atoms only as opposed to considering all possible interactions between atoms in residues (***indicates p < 10−20, Wilcoxon signed-rank test). (B) Scatter plot of PPVs for each protein according to Cα and side-chain center contact identification methods. (C) Histogram of the ratios from the data in (B) indicate that the median percent increase in accuracy is 43%. (D) and (E) As in (B) and (C), but now comparing Cβ and side-chain center contact identification methods. Results show a median percent increase in contact identification accuracy of 13% (Results for two other evolutionary coupling algorithm implementations can be found in Figs. S5 and S6. Further analyses performed with fixed distance cutoffs can be found in Figs. S7 and S8).

We also considered an alternative method for computing contacts: determining structural contacts based on the minimum distance between any two atoms belonging to different two different residues (Ovchinnikov et al., 2017). We implemented two versions of this algorithm, determining the minimum distance between: (i) all heavy atoms within residues and (ii) side-chain heavy atoms only. In each case, and to continue to facilitate a direct comparison between methods, we selected the n shortest distances as contacts where n is the number of contacts identified for each protein via the 8Å distance cutoff using Cα. The resulting PPVs were significantly higher when contacts were defined only according to side-chain atoms as opposed to the complete set of backbone and side-chain atoms (Fig. 4A). We further note that PPVs calculated via side-chain center distances were statistically indistinguishable from PPVs derived from the minimum distance between all heavy atoms within side chains.

All analyses thus far have been performed using a variable distance cutoff to identify true contacts, such that an equal number of true contacts were identified for each structure regardless of the method. This choice was made to facilitate the comparison of PPV values across different contact identification methods. If one contact identification method, for instance, were to identify half the number of true contacts for a given structure compared to another method, the expected PPV would be very different and thus PPV values could not be meaningfully compared without further normalization or re-scaling. However, it would nevertheless be useful to have a uniform distance cutoff to apply to all structures and such a uniform cutoff is more physically realistic and practically useful. We thus repeated our analysis after fixing the Cβ and side-chain center distance cutoffs at 7.6 and 7.5Ås, respectively (values chosen as being close to the median for the distributions displayed in Fig. S1).

For Cα, Cβ, and side-chain center methods, there was no significant difference in the number of contacts identified within each structure (Fig. S7A) and the conclusions of Fig. 4 with regard to PPV remain the same (Fig. S8). However, for comparisons involving the minimum distance between atoms, we found that considering side-chain atoms only resulted in substantially fewer true contacts (using a distance cutoff of 4.5Å between any two heavy atoms to define contacts, Fig. S7A) and the raw PPVs between these two methods cannot be meaningfully compared. By contrast, the average precision score (the weighted mean of PPV across all recall thresholds, similar in spirit to the area under the precision-recall curve) is another measure of accuracy in binary classification tasks and is less sensitive to variability in the number of true positives. We thus compared the average precision scores for all methods using fixed distance cutoffs and found that contacts identified based on side-chain centers resulted in the highest average precision score. Further, considering only the side-chain atoms when looking at the minimum atomic distance resulted in significantly higher scores compared to considering all atoms (Fig. S7B). Taken together, this analysis of fixed distance cutoffs is more physically meaningful than our previous approach that used protein-specific cutoffs to facilitate statistical comparisons, and the results support the finding that evolutionary couplings are largely reflective of side-chain based residue–residue interactions.

Side-chain orientations are important to consider when examining residue–residue interactions

The preceding analyses have shown that for the exact same evolutionary couplings, PPVs are substantially higher when using side-chain based distances to identify true positive intramolecular contacts than when using either Cα or Cβ-based distances. To look more specifically at why these differences were so pronounced, we decided to investigate the orientation of residue–residue pairs identified by the various criteria. Any two residues considered to be in contact can display one of three distinct orientations of their respective side chains: (i) both residues’ side chains may point toward one another with the energetic interactions occurring through side-chain atoms, (ii) one residue’s side chain may point toward the other residue while that residue’s side chain points away, or (iii) both residues’ side chains may point away from one another with energetic interactions occurring between the respective amino acid backbones (Fig. 5A). As we expected, the Cα-based definition yields fractions close to the random expectation for 1AOE: in ∼25% of the cases side chains point toward each other, in another ∼25% of the cases they point away from each other, and in the remaining ∼50% of the cases one side chain points toward the other while the other points away (Fig. 5B). By contrast, side-chain (and to a lesser degree Cβ) based contact definitions enrich for cases where both side chains point toward one another (Figs. 5C and 5D).

Figure 5 Different types of residue–residue interactions are possible.

(A) Two interacting residues may interact via: each residue’s side-chain atoms (type i), the side chain of one residue and the backbone of the other (type ii), or the backbone atoms of each residue (type iii). (B–D) For intramolecular contacts identified in PDB:1AOE, the relative proportion of different interaction types varies according to contact identification method. Shown are contacts identified via Cα atoms (B), Cβ atoms (C), and side-chain centers (D); residue–residue contacts defined via side-chain centers are enriched in type i interactions (blue). (E) For 150 proteins, the fraction of residue–residue pairs where the side chains point toward one-another is highest in contacts defined via side-chain centers (purple). Further, the top ranked evolutionary couplings are progressively enriched in residue–residue pairs where the side chains point toward one another (yellow).

Across all 150 proteins in our dataset, we calculated the fraction of all residue–residue pairs (regardless of distance in 3D space but subject to the same chain distance constraints applied throughout this manuscript) where both side chains point toward one another and found a median of 0.2 (Fig. 5E, “All pairs”). However, this fraction increases progressively when we consider residue–residue pairs identified as true contacts for each protein according to Cα, Cβ, and side-chain centers—illustrating that the trend observed in (Figs. 5B–5D) applies broadly. If instead we consider residue–residue pairs predicted by the top ranked evolutionary couplings, we observe that a large fraction of these couplings are between residues whose side chains point toward one another in the reference protein structure. Additionally, this fraction is highest for the most highly ranked evolutionary couplings (Top 0.25L, where L is the length of the protein) and is substantially higher than the proportion identified by Cα-based distances (Fig. 5E).

This analysis of side-chain orientations highlights that strong evolutionary couplings frequently occur between residues whose side chains point toward one another. Compared to the side-chain center based method, Cα- (and to a lesser extent Cβ-) based contact definitions classify a smaller number of contacts in this orientation and this is likely the cause of decreased predictive accuracies that we observe when using these latter methods to define contacts (Fig. 4).

Discussion

The co-evolutionary patterns of amino acid substitutions can provide important information about protein structures. A number of competing methods are currently in use to detect evolutionary couplings between residues, and the ability to recover true residue–residue contacts has been the primary metric employed to assess performance of various methods. However, true structural contacts are ill-defined and variability in contact definitions can prohibit comparison between the efficacy of different methods as well as obscure the biological interpretation of evolutionary constraints. We have shown here that evolutionary couplings are significantly more accurate at detecting true residue–residue contacts defined via side-chain center distances compared to Cα or Cβ distances. This finding provides important biological insight into protein evolution and epistatic interactions between residues. We posit that although different types of interactions between amino acid residues may stabilize protein structures, evolutionary couplings predominantly represent residues whose contact occurs via interactions between the side-chain atoms of both residues.

While we have shown that using side-chain centers as a reference point to define intramolecular contacts is advantageous compared to Cα or Cβ atoms, it is important to note that the contacts defined via side-chain centers are still likely to be only a rough approximation of reality. Actual contacts between residues within a single structure would ideally be defined via knowledge of the hydrogen bonding, van der Waals forces, and ionic/dipole/hydrophobic interactions that collectively stabilize protein structures (Abdel-Azeim et al., 2014; Doshi et al., 2016; Mercadante, Gräter & Daday, 2018). Defining contacts via these features can be time-consuming and resource intensive and the use of distance-based cutoffs is therefore advantageous under many circumstances. Furthermore, a highly accurate view of residue–residue contacts may still not be particularly useful or informative. For instance, our findings do not suggest that contacts between the backbone atoms of residues do not occur or that they are unimportant for stabilizing protein structures. Rather, we show that contacts between backbone atoms are not likely to be detected by evolutionary coupling analyses. Thus, regardless of the method used for defining residue–residue contacts, different contact types may be of varying importance for different applications.

Amongst distance-based methods, a potential disadvantage of using side-chain centers as reference points is that the geometric center of different amino acid side chains lies at varying absolute distances from the peptide backbone. Here, we have used arbitrary distance cutoffs to define side-chain center contacts (based on identifying an equal number of contacts per protein as an 8Å Cα cutoff). This approach facilitates comparison of different contact identification methods on the same scale, but a more rigorous approach would be preferable. In practice, we have found that a distance cutoff of 7.5Å for side-chain centers works well, and we thus recommend this approach moving forward (Fig. S8), although the physical justification for such a cutoff is unclear. The method of evaluating the minimum distance between side-chain heavy atoms is particularly attractive in this regard, since it permits using a physically realistic and uniform distance cutoff between atoms regardless of amino acid identity or the protein under consideration. While it is possible that analysis of still larger datasets might uncover differences in accuracies between these conceptually similar approaches, our study shows that these methods produce nearly identical results and the choice between them should be based on the tradeoff between simplicity of calculation (side-chain centers) and physical reality (minimum atomic distances between side-chain atoms with a uniform cutoff).

Leveraging homologous sequence information to predict intramolecular protein contacts has long been a goal of structural biologists, but progress toward this goal has been made possible only in recent years with increasing availability of sequence information and the development and application of direct coupling analysis to isolate directly interacting residue–residue correlations from spurious correlation generated by chains of directly coupled residues (Burger & Van Nimwegen, 2008, 2010; Jones et al., 2012; Ekeberg et al., 2013; Seemayer, Gruber & Söding, 2014). However, there are still a number of known limitations to current methods for computing evolutionary couplings as illustrated by the variability in performance that we observe when applying the same basic algorithms to collect and evaluate different homologous protein sets (Fig. 4A). Most evolutionary coupling algorithms require large numbers of sequence homologs to produce accurate predictions (Ovchinnikov et al., 2017); we observed this in our own data, where the number of sequences in a multiple sequence alignment is significantly correlated with the PPV of inferred evolutionary couplings (Fig. S4). However, the size of the multiple sequence alignment explains only a moderate fraction of the variation in accuracies. Features such as the evolutionary relatedness of sequences and the heterogeneity of substitution rates across sites may impose further constraints on the overall identifiability of evolutionary couplings, but it is not clear whether these effects should systematically vary between different protein families (Vorberg, Seemayer & Söding, 2018).

False positive predictions resulting from evolutionary coupling analyses may arise from a combination of different factors including problems with repeat proteins, homo-oligomerization, and structural variation within protein families (Anishchenko et al., 2017). Here, we have shown that another source of false positives in evolutionary coupling analyses may simply be ill-defined true positive contacts. We also observed that accuracy improvements stemming from the use of side-chain centers as a reference point were variable across proteins (Figs. 4C and 4E)—this variability may be related to features of the proteins themselves in terms of their compactness or other physio-chemical properties.

Practically speaking, contact identification methods and evolutionary couplings are increasingly important for a variety of applications. Frequently, evolutionary couplings are combined with a variety of other features and used as input for machine learning and associated neural network-based algorithms to predict structural and functional properties of proteins (Cheng & Baldi, 2007; Tegge et al., 2009; Di Lena, Nagata & Baldi, 2012; Jones et al., 2015; Michel et al., 2017; Xiong, Zeng & Gong, 2017; Stahl, Schneider & Brock, 2017; He et al., 2017; Wang et al., 2017; Riesselman, Ingraham & Marks, 2018; Liu et al., 2018; Wozniak et al., 2018; Jones & Kandathil, 2018; Adhikari, Hou & Cheng, 2018; Hanson et al., 2018). Our analysis suggests that there may be biases in the training data—essential to supervised learning techniques—owing to the method used to define true positive contacts. Given that raw evolutionary couplings more accurately predict contacts identified by side-chain centers, we speculate that the accuracy of supervised learning algorithms may be similarly improved by training and testing on contacts identified via side-chain distances. Different contact types can be classified according to which atoms interact between residue–residue pairs (Fig. 5A), and it is possible that the accuracy of supervised approaches in particular may be improved by separating different types of residue–residue contacts according to their atomic interactions, training separate models to detect each type, and integrating the results.

Conclusions

Sequence based co-evolutionary methods are a powerful tool for studying structural and functional constraints on protein families, and the accuracy of these methods is frequently benchmarked by their ability to predict residue–residue contacts. Here, we show that there are numerous ways to summarize a given protein structure as a set of residue–residue contacts and that the choice of how to do this mapping has important consequences for downstream applications. We find that the predictions of evolutionary coupling algorithms are substantially more accurate when predicting residue–residue contacts defined via their side chains, highlighting the important role that side-chain interactions play in governing epistasis and protein evolution. Based on these results, we recommend defining residue–residue contacts via either side-chain centers (as a fixed reference point) or the minimum atomic distances between side-chain atoms for applications moving forward.

Supplemental Information

Supplemental Information 1 Figure S1: Contact distance thresholds.

To directly compare contact identification metrics, contacts for a given protein were first defined according to Cα atoms with an 8 Å distance cutoff. For Cβ (left) and side-chain center (right), an equal number of putative contacts were then identified regardless of their distances. Histograms show the relevant distance cutoffs that ensure an equal number of putative contacts for each metric for the set of 150 proteins.

Click here for additional data file.

Supplemental Information 2 Figure S2: Contact differences between Cβ and side-chain center based methods.

Similar to Figure 2; here, we compare residue–residue distances defined via Cβ and side-chain centers. (A) For PDB:1AOE, Cβ and side-chain center contacts were defined according to 7.5 A and 7.33 A distance cutoffs, respectively. (B) Distribution of Spearman’s correlation coefficient values (ρ) between residue–residue distances for 150 different proteins. (C) Distribution of the percent agreement for contact definitions for the same set of proteins.

Click here for additional data file.

Supplemental Information 3 Figure S3: Comparing Cβ and side-chain center based contact definitions in simulated data.

Similar to Figure 3; here, we compare Cβ and side-chain center contact definitions. (A) For each of 5 separate selection strengths (colored lines), we ran simulations until a number of mutations totaling 10 times the length of the protein were accumulated per replicate. We varied the number of independent replicate sequences (x-axis) that were used as input for evolutionary coupling analysis, and found that couplings fail to fully recover Cβ defined contacts for PDB:1AOE. (B) By contrast, contacts defined via side-chain centers are near-perfectly recovered across a range of simulation parameters. (C) and (D) Similar to parts (A) and (B), but along the x-axis we now show results from simulations where a different number of accepted mutations were accumulated per sequence. We fixed the number of replicate sequences that were simulated—and used for evolutionary coupling analysis—at 3,000 for each of these data points. (note: side-chain center plots—(B) and (D)—are the same as in Figure 3 and are included here only for comparison.).

Click here for additional data file.

Supplemental Information 4 Figure S4: PPV as a function of alignment size.

PPVs for all three different contact identification methods (y-axes) are strongly correlated with the number of sequences in the empirical sequence alignments used as input for calculating evolutionary couplings. PPV data here is the same data from Figure 4.

Click here for additional data file.

Supplemental Information 5 Figure S5: Empirical results using the PSICOV method.

Similar to Figure 4; here, we use the PSICOV method to calculate evolutionary couplings. (A) For a diverse set of 150 proteins, the PPV of the top L/2 evolutionary coupling scores—derived from empirical sequence alignments—is progressively higher when intramolecular contacts are defined according to Cα atoms, Cβ atoms, and side-chain centers. Similarly, PPVs are higher when computing contacts based on side-chain atoms only as opposed to considering all possible interactions between atoms in residues. (*** indicates p < 10−20, Wilcoxon signed-rank test) (B) Scatter plot of PPVs for each protein according to Cα and side-chain center contact identification methods. (C) Histogram of the ratios from the data in (B) to estimate the effect size. (D) and (E) As in (B) and (C), comparing Cβ and side-chain center contact identification methods.

Click here for additional data file.

Supplemental Information 6 Figure S6: Empirical results using the PLMC method.

Similar to Figure 4; here, we use the PLMC method to calculate evolutionary couplings. (A) For a diverse set of 150 proteins, the PPV of the top L/2 evolutionary coupling scores—derived from empirical sequence alignments—is progressively higher when intramolecular contacts are defined according to Cα atoms, Cβ atoms, and side-chain centers. Similarly, PPVs are higher when computing contacts based on side-chain atoms only as opposed to considering all possible interactions between atoms in residues. (*** indicates p < 10−20, Wilcoxon signed-rank test) (B) Scatterplot of PPVs for each protein according to Cα and side-chain center contact identification methods. (C) Histogram of the ratios from the data in (B) to estimate the effect size. (D) and (E) As in (B) and (C), comparing Cβ and side-chain center contact identification methods.

Click here for additional data file.

Supplemental Information 7 Figure S7: Number of contacts and Average Precision Score using fixed distance cutoffs.

Using a fixed distance cutoff to determine contacts for each method illustrates heterogeneity in contact numbers and prediction accuracy. (A) The number of contacts defined in each of 150 proteins according to the method and the distance cutoff (in angstroms). (*** indicates p < 10−20, Wilcoxon signed-rank test; n.s.indicates p > 0.05). (B) Due in particular to the large difference in the number of contacts identified by the ‘Side-chain atoms min. distance’ method, we repeated the results of Fig. 4 using fixed cutoffs (as indicated in panel A) and the Average Precision Score as a metric of accuracy, which is more comparable than PPV across sets with varying numbers of true positives. (*** indicates p < 10−20, Wilcoxon signed-rank test).

Click here for additional data file.

Supplemental Information 8 Figure S8: Empirical results using fixed distance cutoffs for contact determination.

Similar to Figure 4; here, we show results when using a fixed distance cutoff to define contacts individually for the three separate methods: Cα (8A), Cβ (7.6A), side-chain center (7.5A). (A) For a diverse set of 150 proteins, the PPV of the top L/2 evolutionary coupling scores—derived from empirical sequence alignments—is progressively higher when intramolecular contacts are defined according to Cα atoms, Cβ atoms, and side-chain centers. (*** indicates p < 10−20, Wilcoxon signed-rank test) (B) Scatter plot of PPVs for each protein according to Cα and side-chain center contact identification methods. (C) Histogram of the ratios from the data in (B) to estimate the effect size. (D) and (E) As in (B) and (C), comparing Cβ and side-chain center contact identification methods.

Click here for additional data file.

The authors acknowledge valuable feedback and support from members of the Wilke lab.

Additional Information and Declarations

Competing Interests

Author Contributions

Data Availability

Claus O. Wilke is an Academic Editor for PeerJ.

Adam J. Hockenberry conceived and designed the experiments, performed the experiments, analyzed the data, prepared figures and/or tables, authored or reviewed drafts of the paper, approved the final draft.

Claus O. Wilke conceived and designed the experiments, contributed reagents/materials/analysis tools, prepared figures and/or tables, authored or reviewed drafts of the paper, approved the final draft.

The following information was supplied regarding data availability:

All data has been permanently archived at Zenodo:

https://dx.doi.org/10.5281/zenodo.2552779.

Code used for all analyses is available on GitHub:

https://github.com/adamhockenberry/side-chain-couplings.

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
