# Peer review of "Evolutionary couplings detect side-chain interactions"

_PeerJ, doi:10.7717/peerj.7280_

## Round 0.1 · original submission · Minor Revisions

The reviewers have a number of comments, mainly editorial. Reviewer No. 1 has a suggestion about extending the analysis to tools based on contact networks; I leave it to the authors whether to follow this suggestion, but in any case such tools should be mentioned in the review or discussion section.

Reviewer 1 ·

Basic reporting

The manuscript presents a somewhat refreshing perspective to the question of a high current relevance, namely, how to measure the performance of contact-prediction sequence-based methods. The authors show a broad knowledge of the field, give a sufficient background, and generally the manuscript is well-written and comprehensible.

Experimental design

The experimental design is sound to the problem stated. My only concern is the introduction of the minimum distance between heavy atoms as a contact measure very late into the results section. The authors (quite rightfully) state that this measure is perhaps physically most relevant, so in my opinion it should feature prominently throughout the results section right from the beginning. I would recommend restructuring this section so that it is compared to C-alpha-based distances and geometric center-based distances in Figs. 2-5. Then I would appreciate a comment on why center-based distance might be preferred (computational efficiency, I presume?).

Another way to detect contacts that could be interesting in this context are resudue interaction networks (e.g. https://academic.oup.com/nar/article/44/W1/W367/2499329, https://bmcproc.biomedcentral.com/articles/10.1186/1753-6561-8-S2-S2). The authors might want to look into these tools.

Minor comment: is residue center a center of mass or geometric center? I understand that they are spatially close, but this should be clearly stated.

Another minor comment: It should be clearly explained how the distance cutoffs are chosen. It is mentioned somewhere in passing that they are chosen to pick the same number of contacts as the 8A cutoff for C-alpha's, however, I would appreciate if that was stated somewhere around the reference to Fig. S1.

Third minor comment: In the figure legend to Fig. S1, 150 proteins are mentioned, which have not been introduced to this point. Please fix the legend.

Validity of the findings

The obtained results are perfectly sound and supported by the data.

Additional comments

I have some additional minor comments to the presentation:

1. SI Fig. S7 seems important to me and should be oved to the main text.
2. Neural networks (mentioned on p. 9) are a type of machine-learning algorithms. Please fix the wording.

·

Basic reporting

The manuscript is written unambiguously and organized well. The introduction is detailed, relevant and well referenced. Figures and tables are comprehensive and helpful.

There are some minor comments:
1) Line 29: A reader may expect here supporting references to recent relevant reviews on X-ray crystallography, cryo-electron microscopy, and other listed methods.
2) Line 28: “...primary amino acid sequence...”. Here, the word “primary” has to be omitted (there are no secondary or tertiary sequences). Or, alternatively, the authors can use the term “primary structure”.
3) Lines 93, 137, 195, 323: “...primary chain...”. Again, the word “primary” has to be omitted (there is no secondary or tertiary chains).
4) Line 65: “... can be chosen to use as a reference point … “ The words “to use” can be omitted.
5) Line 200: “ … large fraction of the these couplings ...” The word “the” should be removed.
6) Legend to SI Fig. S3, line 5: “... and found that couplings fail to fully recover C-alpha defined contacts...” Should be “C-beta” instead of “C-alpha”.

Experimental design

The experimental design was clear. The question is clearly stated and the authors answer it unambiguously.

The authors investigate the relationship between the residue-residue covariation revealed by couplings analysis methods and the type of defining the contacts between residues. In other words, they ask the question, what type of contacts are found by coupling analysis methods? The answer is that these are contacts that are formed between geometric-center of side-chains. This is expected because it is side-chains (not C-alpha atoms) that mutate coordinately in the course of evolution. But the value of the study is that the authors confirmed this expectation explicitly and quantified the found effect.

The only small concern is a requirement that the number of contacts made by side-chain centers was the same as the number of C-alpha contacts. As a result, the authors use different values of the threshold for different proteins, which looks artificial from the physical point of view (lines 89-92). If the way of calculation of the contacts does not influence the results, why then not to show results for a more general method of detecting contacts in 3D structure, with the same threshold for all proteins? I don't see a problem that the number of C-beta or geometric-center contacts will be different from C-alpha ones. If the authors want to compare histograms they can normalize the histograms before comparison or compare cumulative distribution like in Kolmogorov-Smirnov test.

Validity of the findings

Data are robust and controlled. Conclusions are well stated and relevant.

Additional comments

None

---

## Round 0.2 · accepted · Accept

The reviewers' concerns have been adequately addressed, and the manuscript has become both stronger scientifically and better readable.